# Regulation of Oxygen Homeostasis at the Intestinal Epithelial Barrier Site

**DOI:** 10.3390/ijms22179170

**Published:** 2021-08-25

**Authors:** Špela Konjar, Miha Pavšič, Marc Veldhoen

**Affiliations:** 1Department of Chemistry and Biochemistry, Faculty of Chemistry and Chemical Technology, University of Ljubljana, SI-1000 Ljubljana, Slovenia; miha.pavsic@fkkt.uni-lj.si; 2Instituto de Medicina Molecular, João Lobo Antunes, Faculdade de Medicina de Lisboa, 1649-028 Lisbon, Portugal; marc.veldhoen@medicina.ulisboa.pt

**Keywords:** oxygen, mitochondria, hypoxia, microbiota, IBD

## Abstract

The unique biology of the intestinal epithelial barrier is linked to a low baseline oxygen pressure (pO_2_), characterised by a high rate of metabolites circulating through the intestinal blood and the presence of a steep oxygen gradient across the epithelial surface. These characteristics require tight regulation of oxygen homeostasis, achieved in part by hypoxia-inducible factor (HIF)-dependent signalling. Furthermore, intestinal epithelial cells (IEC) possess metabolic identities that are reflected in changes in mitochondrial function. In recent years, it has become widely accepted that oxygen metabolism is key to homeostasis at the mucosae. In addition, the gut has a vast and diverse microbial population, the microbiota. Microbiome–gut communication represents a dynamic exchange of mediators produced by bacterial and intestinal metabolism. The microbiome contributes to the maintenance of the hypoxic environment, which is critical for nutrient absorption, intestinal barrier function, and innate and/or adaptive immune responses in the gastrointestinal tract. In this review, we focus on oxygen homeostasis at the epithelial barrier site, how it is regulated by hypoxia and the microbiome, and how oxygen homeostasis at the epithelium is regulated in health and disease.

## 1. Introduction

The primary roles of the intestinal epithelial barrier are absorption of nutrients, fluid homeostasis, removal of waste products, and maintenance of tolerance to antigens [1]. Intestinal epithelial cells (IEC) are located at the interface of the nutrient and microbial ecosystem. The monolayer of IECs is a key player in maintaining metabolic and immune homeostasis [2]. The epithelial layer can be completely renewed in 3–5 days and serves as a protective mechanism against injury and infection [3]. Renewal of the epithelial layer of the IEC requires a specific population of IEC called intestinal stem cells (ISCs) [4].

The epithelium, covered with mucous, is supplied by an abundant vascular system. Even a minor disturbance in blood flow can cause a substantial decrease in O_2_ supply (hypoxia) to the epithelia. The small and large intestine can quickly adapt to the widely fluctuating O_2_ levels in blood perfusion [5].

At the baseline of epithelial cells lining the mucosa, relatively low pO_2_ exists, which is referred as “physiological hypoxia” [6]. Cells can adapt to oxygen supply and demand under hypoxia, demonstrating that cells must have oxygen sensing mechanisms regardless of their temporary biological status. Mitochondrial signals represented by reactive oxygen species (ROS) and metabolites can mimic acute and chronic hypoxia responses [7]. Rapid acute responses occur as a result of ROS-induced or metabolite-induced Ca^2+^ accumulation in specialized oxygen-sensitive cells. Long-term hypoxia sensing is possible through ROS and metabolites due to hypoxia-inducing factor (HIF) activation [7].

The epithelial cells are overlaid by a mucus surface, which is an important barrier to the flow of antigens from the lumen. The mucosal surface harbours trillions of microbes, including bacteria, fungi, and viruses. Collectively, they produce an array of fuel sources and signalling molecules such as short-chain fatty acids (SCFAs), including butyrate, propionate, and acetate. Microbially derived SCFAs, particularly butyrate, stimulate epithelial metabolism, alter gene expression [8], and increase epithelial O_2_ consumption to a level that cells perceive as metabolic hypoxia, which leads to the stabilization of the transcription factor HIF [9].

In turn, intestinal oxygenation directly shapes the composition of gut microbial communities and oxidative changes in intestinal inflammation and may underlie the characteristic dysbiosis related to the microbiota in patients with IBD [10]. Here, we will discuss how the intestinal epithelium functionally adapts to a low O_2_ pressure (pO_2_) environment, how crosstalk between the microbiota and hypoxia functions, and how oxygen metabolism at the epithelial barrier is affected in health and disease.

## 2. Oxygen Scenery at the Intestinal Epithelial Barrier

At the intestinal barrier, the unique oxygen profile is governed by two main factors—by oxygen metabolism of epithelial and sub-epithelial cells during food digestion and nutrient absorption, and by intestinal microbiota within the lumen. For example, the energy expended during digestion and absorption can account to up to 10% of total energy expenditure, measured as dietary induced thermogenesis [11]. During the metabolic processes of digestion and absorption, gastrointestinal oxygen consumption is increased disproportionately compared to the gastrointestinal blood flow, resulting in hypoxia [12]. Adding to the complexity, the gut microbiome is composed mainly of anaerobic bacteria which must, however, be able to quench the reactive oxygen species produced during aerobic host metabolism [13]. At the same time, the gut microbiome is critically involved in the maintenance of hypoxic conditions within the lumen, which is required for normal intestinal functions including nutrient absorption [14].

The characteristic features of the digestive tract are the longitudinal and the radial steep spatial oxygen gradient (Figure 1). Longitudinally—from the small intestine to the colon—the oxygen level drops (Figure 2) [15,16], and a negative gradient is also observed from lamina propria towards the lumen. To illustrate this, in mice, pO_2_ drops from 58 mmHg in the stomach to 32 mmHg in the duodenum, through 11 mmHg in the ascending column and to 3 mmHg in the sigmoid column (all values measured using electron paramagnetic resonance, EPR) [16]. In contrast, the pO_2_ at the sea level is 140–150 mmHg (corresponding to ~21% O_2_), and at the lung alveoli, it is still high at ~105 mmHg [17]. Considering the short distance, the radial gradient is even steeper—using the phosphorescence quenching method, it was shown that in mice, the pO_2_ drops from 40 mmHg in the subepithelial tissue to below 1 mmHg in the lumen over a distance similar to the thickness of the intestinal wall (~300 µm) [10]. While the values obtained using different methods are not directly comparable, they nevertheless depict the striking differences in oxygen levels at various points of the gastrointestinal tract.

As noted, several methods are available for measurement of pO_2_ in tissues of living organisms (Table 1), and due to different experimental approaches, the pO_2_ values obtained using these methods may not be directly comparable. The first instrument that made in vivo measurements possible was the Clark electrode, which, at its core, is a platinum electrode at the surface of which oxygen is electrochemically reduced and the generated current is measured. Despite several drawbacks, it still remains widely in use, particularly due to its simplicity. While newer methods based on nuclear or electron magnetic resonance provide unprecedented possibility to obtain 3D oxygen profiles in tissues or even the whole body, their use is limited due to the expensive and complex equipment required. The advantages and disadvantages of the main methods in use today are collected in Table 1.

Complementary to these methods to determine pO_2_ values over a wide range, there are several methods that generally do not give exact pO_2_ values, but they do provide a means to visualize oxygen gradients and detect low oxygen conditions. For example, exogenous hypoxia markers added before tissue biopsy are in use both in animal and experimental medicine for ex vivo hypoxia imaging [25]. Common markers are nitroimidazole dyes—e.g., pimonidazole and pentafluoropropyl—which are in hypoxic cells (pO_2_ < 10 mmHg) reduced and retained due to the formation of a dye–protein adduct. One of the standard methods for immunohistochemical hypoxia detection is also pimonidazole immunofluorescence, utilizing anti-pimonidazole antibody conjugated to a fluorescent marker compound. A recent study using brain tumour models showed good correlation between pimonidazole-detected hypoxia and positron emission tomography (PET) measurements, thus bringing the latter non-invasive method closer to clinical application [26]. When comparing different oxygen level measurement methods, it is also important to consider whether the measurement is limited to the lumen. For example, the classical method using the Clarke electrode allows for luminal hypoxia measurement, while the pimonidazole method also images intracellular hypoxia. Another consideration is that some probes could be endocytosed such as in the case of the lithium naphtalocyanine EPR probe, making luminal-only measurements problematic [27]

Alongside direct measurements using electrodes and indirect measurements of hypoxia using exogenous markers, there are also endogenous hypoxia (bio)markers that can be detected using immunohistochemistry. Of these, the most important is HIF-1α, which is described in detail in the following sections. Other endogenous markers are carbonic anhydrase IX (CA IX) and glucose transporter 1 (GLUT-1), plus some less-specific markers such as osteopontin (OPN) and vascular endothelial growth factor (VEGF) [28].

## 3. Oxygen Sensing through Hypoxia and ROS at the Mitochondria Site

The intestinal epithelium is constantly exposed to low-oxygen conditions. The hypoxic environment can result in altered membrane protein distribution and membrane lipid composition with an increase in saturated fatty acids content, while the amounts of phospholipids and cholesterols remain similar [29]. Hypoxia is thought to have a protective role in broader aspects such as infection and inflammation [30,31]. Therefore, intestinal epithelium is a prototypical environment in which adaptations to hypoxia are key [32]. One of the most important regulatory mechanisms of oxygen involve HIFs [33] and reactive oxygen species (ROS) produced in mitochondria.

HIF is a global regulator of oxygen homeostasis and enables adaptation to oxygen levels in numerous cell types, including intestinal epithelial cells [34]. It is a heterodimer of a constitutively expressed beta subunit HIF-1β, also known as aryl hydrocarbon receptor nuclear translocator (ARNT), and an alpha subunit HIF-1α, which is constitutively translated but rapidly degraded under physiological oxygen conditions (normoxia). In the absence of oxygen, HIF-1α binds the constitutively expressed partner HIF-1β and thereby stabilizes the heterodimer, which is consequently recruited to HIF response elements (HREs) present in the promoter region of HIF target genes where its transcription factor activity is mediated by its basic helix-loop-helix-PAS motif within HIF-1α (Figure 1) [35].

The critical component of the HIF-mediated oxygen-sensing mechanism is the post-translational modification of HIF-1α. During normoxia in the richly vascularised subepithelial mucosa in the crypts of the intestine, the HIF-1α subunits are hydroxylated by a group of 2-oxoglutarate-dependent dioxygenase enzymes named prolyl hydroxylases 1, 2 and 3 (PHD 1, 2, 3) [7]. The hydroxylation occurs in the HIF-1α oxygen-dependent degradation domain and targets HIF-1α for ubiquitination by tumour suppressor protein Von Hippel–Lindau (VHL) and, in turn, for proteasomal degradation (Figure 1). PHD 1–3 also can hydroxylate HIF subunits to prevent transcriptional activation. On the contrary, under hypoxic conditions in anaerobic lumen of the intestine, the dioxygenase activity of PHD is inhibited, non-hydroxylated HIF-1α protein accumulates and is recruited to the nucleus with other HIF transcription factors. Collectively, they trigger a programme of gene expression that controls cell fate and activates alternative mechanisms of energy production. This also leads to the expression of genes crucial for the mucosal barrier function, such as junction proteins, antimicrobial peptides, and mucus [36,37]. Hypoxia increases the expression of hundreds of mRNAs and decreases the expression of a similar number of genes in a HIF-dependent manner [38]. It is estimated that after hypoxic activation, HIFs can bind approximately 500 different sites in genes as a transcription factor depending on cell type [39,40,41]. Furthermore, HIF-1α and/or HIF-2α are involved in the induction of genes encoding enzymes involved in glycolytic, carbohydrate, fatty acid, mitochondrial and peroxisome metabolism, which has been discussed elsewhere [42].

The other major regulatory mechanism involves mitochondria, which are responsible for most of the oxygen consumption in cells. Recently, it has been shown that mitochondria can serve as signalling organelles required for numerus cytosolic signalling pathways [43]. This signalling in mitochondria can be through ROS or through the release of tricarboxylic acid cycle (TCA) metabolites [7,44]. For example, the TCA-metabolite citrate can be transferred to the nucleus to create cytosolic acetyl CoA that is required for protein acetylation, which can alter signalling pathways via post-translational modifications and gene expression via an altered epigenetic status [45].

The utilisation of oxygen in mitochondria occurs mainly at the electron transfer chain (ETC) complex IV (cytochrome C oxidase). From oxygen, the universal unit of cellular energy called ATP is generated in a controlled cascade. At the ETC, O_2_ serves as the terminal electron acceptor of the electron transport chain to form H_2_O in the process of oxidative phosphorylation. Along the ETC, there are many possibilities where electron leakage can occur and consequently, this has implications for mitochondrial signalling [46]. When an electron is lost, a single electron is picked up by O_2_; combined, they can form a superoxide radical, incompletely reduced O_2_ radical, hydrogen peroxide, or hydroxyl radical, all commonly referred to as ROS. The mitochondrial electron transport chain (mETC) is a major source of mitochondrial ROS, where the predominant reactive species is superoxide. mETC complex I and II are the major producers of superoxide and they release it into the mitochondrial matrix and intermembrane space [4].

Tissue cells and organs are exposed to a wide range of O_2_ levels. Insufficient or excessive levels of O_2_ result in inappropriate levels of ROS, leading to altered oxidation of lipids, proteins, and nucleic acids, which can lead to cell death or dysfunction [47]. Excessive mtROS that are not eliminated by mitophagy can lead to multiple cellular dysfunctions [48]. For example intestinal stem cells (ISCs) in intestinal organoids that were not capable of eliminating excessive mtROS with mitophagy/autophagy have lower self-renewal potential, leading to their reduced number [49].

The small intestine and colon mainly consume ATP released from oxidative phosphorylation. Epithelial oxygen consumption and regulation is an important determinant of oxygen balance at the interface between host and environment. Therefore, both the delivery and consumption of O_2_ are tightly regulated, with many different molecular mechanisms controlling O_2_ levels within cells and tissues to maintain oxygen homeostasis. Mitochondria ETC in the body range functionally from ambient O_2_ conditions (21% O_2_), through hyperoxia and hypoxia, to near anoxia (approximately 0.5% O_2_). Any deviation from these conditions (either up or down in pO_2_) can result in pathology. Mitochondrial ROS signals are required for various cellular functions and their amount is connected to the activities of the mETC, metabolic enzymes and activation of AMPK, thiol phosphatase, caspase, kinase, proteasome as well as transcriptional activity [4,50]. ROS may have additional effects on stem cell differentiation, proliferation, and adaptation to stressors including hypoxia [4,32]. Various genetic and antioxidant methods have been used to demonstrate the necessity of increasing ROS for oxygen sensing [29].

Mitochondrial ROS are crucial for the activation of HIF [51]; the exact mechanism of ROS-mediated HIF-1α stabilisation is still unknown. There are debates in the literature on whether mitochondria ROS increase or decrease during hypoxia discussed elsewhere [52,53]. However, various genetic and antioxidant methods have been used to demonstrate the necessity for detecting ROS for oxygen sensing [54]. Hypoxia has been shown to increase mitochondrial ROS generation at the complex III and increase HIF-1α protein accumulation [55]. In addition, mitochondrial intermediates used in the TCA for energy production, such as succinate and fumarate, have been shown to increase ROS production and indirectly activate HIF-1α protein [14,51].

Diminished oxygen levels at the intestinal barrier site cause stress and destabilize cells at the intestinal barrier site. This environment adapts systemically, which can include changes in ventilation, cardiac output, blood vessel growth and circulating blood cells. Regulation of low oxygen could go through the hypoxia mechanism directed by HIF enzymes. HIF regulation stabilizes low oxygen availability and controls the expression of a multitude of genes, including those involved in cell survival, angiogenesis, glycolysis and invasion/metastasis. Hypoxic reprogramming of metabolism can also be associated with adaptation to changes in ROS production, which goes along with other mitochondrial changes during hypoxia [14,52]. All these changes can lead to adaptations of cells to protein synthesis, energy metabolism, mitochondrial respiration, lipid and carbon metabolism as well as nutrient acquisition.

## 4. Microbiota and Oxygen Consumption

The intestines of newborns are better oxygenated compared to those of adults. The higher oxygen content in the intestinal tract of newborns favours the occurrence of facultative anaerobes such as *enterobacteria*, *enterococci*, and *streptococci*. These facultative anaerobes consume the available oxygen, create an anaerobic microenvironment in the intestine and facilitate the establishment of obligate anaerobes such as *Bifidobacterium, Clostridium* and *Bacteroides*. This shows that oxygen levels can influence the composition of the gut microbiota [56]. The number and heterogeneity of commensals increases along the longitudinal axis of the intestine from the small intestine to the colon and also between tip and base of villus of the intestinal epithelium [14] (Figure 1 and Figure 2). In addition, this leads to increased amounts of oxygen used by aerobic bacteria near the epithelium, thereby enabling an increase in aerotolerant microbes at the interface between the epithelium and the lumen, leaving the central part of the lumen deoxygenated [14].

The importance of the microbiota in regulating intestinal pO_2_ was shown using antibiotic administration, which affect both the intestinal oxygen gradient and microbial composition [56]. Similar observations have been made in germ-free mice, which also show higher pO_2_ in the intestinal epithelium compared to controls. Butyrate supplementation of antibiotic-treated mice physiologically restores levels of low pO_2_ and hypoxia-dependent signalling. However, protection of the intestinal epithelial barrier by butyrate did not occur in cells lacking HIF [9]. These results indicate that the butyrate–HIF axis is an important pathway in host–microbe interactions. Another metabolic deficiency reported in GF mice is that colonic cells show significantly reduced levels of ATP and NADH/NAD^+^ in addition to decreased capacity for oxidative phosphorylation [57].

Depletion of butyrate-producing bacteria by antibiotics decreased the expression of peroxisome proliferator-activated receptor-γ (PPARγ), a receptor mediating butyrate oxidation. The reduction in PPARγ increased oxygen availability in the gut lumen, which promotes the colonialization with aerobic *Enterobacteriaeae* [58,59].

These observations suggest that a viable gut microbiota that enables stable oxygen gradient is important for preserving intestinal homeostasis and, consequently, resistance to pathogen colonization. Decreasing normal microbiota with antibiotic manipulation could lead to imbalanced oxygen homeostasis (increase in O_2_ levels, growth of aerobic bacteria) that can lead to pathological conditions and epithelial damage [56,58,60].

## 5. Microbiota and Its Metabolites

The gut microbiota is composed of more than 100 trillion microorganisms [61], consisting of approximately 160 species [62]. A major role of the microbiota is the breakdown of nutrients from food and the controlled regulation of intestinal and systemic immune responses. In addition to assisting in digestion, the microbiota produces a number of signalling molecules and benefits the host through the local synthesis of SCFA. Bacteria present in the microbiota can also produce various gaseous molecules such as nitric oxide (NO), carbon monoxide (CO), hydrogen sulphide (H_2_S), and ammonia (NH_3_) that interact with HIF [14]. These complex interactions between microbiota, epithelial barrier and nutrients are also dependent on oxygen homeostasis at the epithelial barrier site. Microbiota metabolites influence oxidative phosphorylation [63], nuclear receptors [64] and other functions related to the metabolism at the epithelial barrier site of intestine [59].

Facultative anaerobic bacteria are the predominant bacterial species in the microbiota and they decrease the O_2_ content in the environment; therefore, the gut in general is a highly hypoxic tissue compared to other tissues. In the intestine, anaerobic bacteria ferment and extract energy from other polysaccharides such as dietary fibre and starch and synthesize the by-product SCFA consisting of primary N-butyrate, propionate and acetate, which are present in high concentrations in the lumen [65]. A high fibre diet results in the production of 400–600 mmol of SCFA in the cecum per day; these account for ~10% of human caloric requirements [66].

Primarily, SCFA are involved as energy substrates in oxidative phosphorylation, where oxygen is consumed for the production of ATP. The type and amount of SCFAs produced depends on the composition of the gut microbiota and the metabolic interaction between microbiota and ingested macro- and micronutrients [66]. SCFAs are also mediators of metabolism in mitochondria as they serve as ligands for free fatty acid receptors 2 and 3 (FFAR2, FFAR3), also known as G-coupled receptor protein 43 (GRP43) and GRP41, respectively, which regulate glucose and fatty acid metabolism [66]. Small intestinal epithelial cells have a preference to utilise glucose and glutamine [67], whereas mature colonic epithelia mainly generate energy by oxidation of SCFA such as butyrate, which may render the mucosal surface more hypoxic. Increased SCFA correlate with reduced lipolysis and adipogenesis of adipocytes [68,69] and also inhibit insulin-stimulated lipid accumulation in adipocytes via FFAR2 signalling, resulting in more responsive adipocytes associated with reduced inflammatory infiltrate in the adipose [10,69,70].

In addition to SCFA’s role as an energy substrate mentioned above, SCFA also have several immunomodulatory effects [37,71]. Treatments where they increase the intestinal concentrations of SCFA are effective in reducing tissue damage and improving immune response. In this way, SCFA can improve the host response to inflammatory and infection stimuli [72,73]. There is increasing evidence supporting a homeostatic role for SCFA in colonic inflammation in both strengthening the intestinal barrier and promoting healing of colitis [74]; this is discussed in more detail in the next section.

Butyrate is one of the predominant SCFAs in the colon and can reach luminal concentrations of up to 30 mM. It is a metabolic substrate for colonic epithelial cells, with up to 30% of energy in the healthy colon derived from butyrate [63]. An increase in butyrate production has been observed in mice harbouring the microorganisms *Roseburia intestinalis*, *Eubacterium rectale*, and *Clostridium symbiosum* that were on low-fat, high-fibre diets [63]. In contrast, in mice that were on high-fat, low-fibre diet and with consequently low butyrate production, there was an increase in CD3^+^ intraepithelial lymphocytes and CD68^+^ lamina propria macrophages, suggesting increased inflammation in the absence of fibre degradation and SCFA production [75]. Propionate and butyrate are established as activators of nuclear receptors such as PPARγ [76,77], which, after activation, mediate mitochondrial β-oxidation of fatty acids [57,78] and, as mentioned before, receptor mediated butyrate oxidation [58]. By fuelling oxidative phosphorylation, butyrate is critical for initiation and to sustain a hypoxic/HIF gradient in the intestine [9,14]. Administration of butyrate to germ-free mice rescues mitochondrial respiration in colonocytes and prevents them from undergoing autophagy [57].

All these studies show that butyrate has many beneficial roles in the intestine [63,75,76]. However, butyrate can have deleterious properties, sometimes referred to as the “butyrate paradox” that was observed in studies looking at the impact of butyrate on colonic stem cells versus colonocytes [79]. In these studies, butyrate was identified as a potent suppressor of stem cell proliferation and inhibitor of histone deacetylase (HDAC). This results in chromatin remodelling and changes in gene expression. On the contrary, colonocytes were resistant to butyrate-mediated HDAC inhibition with no significant alteration in gene expression. Colonocytes also tolerated higher doses of butyrate compared to stem cells [80].

Microbial ligands can be recognised by pattern recognition receptors (PRR); an example is the activation of PPR-related transcription factors in IEC, such as via *Bacteroides vulgatus*-mediated NF-κB signalling [81]. In addition to PRRs, luminal metabolites from food, gut microbiota and digestive fluids (e.g., dietary lipids, SCFA, bile acids and many more) are sensed via a variety of host-expressed metabolic receptors. Metabolic activities of the microbiome include influencing the tryptophan-related aryl hydrocarbon receptor. The small intestine of wild-type mice fed with a diet depleted of AhR ligands harbours lower levels of *Firmicutes* and higher levels of phyla *Bacteroidetes* than mice fed with a diet containing AhR ligands [82]. The intestine of mice deficient in AhR also has increased levels of phyla *Bacteroidetes* and these mice are very susceptible to DSS-colitis [83]. AhR signalling is essential for normal gut immune function and is triggered by plant compounds that are converted to AhR ligands in the gut environment [83].

Metabolites derived from microbiota can serve as metabolic sensors with several important roles at the intestinal epithelial barrier. They can serve as fuel in metabolic processes, have an impact on hypoxic regulation, have immunomodulatory effects, activate nuclear receptors, activate vitamin D receptors, and consequently, affect oxygen homeostasis. Organic synthesis of these metabolites is a good opportunity to manipulate gut microflora for medical purposes and to carry out further research in the field of intestinal microbiota. Although, it is likely that the implication of artificial versions of these metabolites will have a lower impact on the organism as metabolites produced naturally by healthy viable microbiota. However, much research in this direction needs to be undertaken to elucidate the role of metabolites in the biology of epithelial cells and immune cells at the intestinal barrier [64,84].

## 6. Oxygen Sensing at the Epithelial Barrier in IBD

Oxygen homeostasis in healthy intestines is assured by dynamic and rapid fluctuation in cellular oxygen tension. In inflammatory bowel disease (IBD), this cellular oxygen tension is dysregulated [40]. IBDs are chronic, immunologically mediated diseases of the gastrointestinal tract. The current understanding of IBD pathogenesis assumes that in genetically predisposed individuals, an improperly regulated interaction between the microbiota and the mucosal immune system leads to an inappropriate inflammatory response and that disease recurrence is triggered by environmental factors that remain to be fully characterized (Figure 3) [85]. The sensors that detect fluctuating oxygen levels appear in normal and disease states. They are an important link between metabolism and chromatin regulation; therefore, it is warranted to further address how oxygen-regulated metabolic adaptations affect the epigenome and how this impacts the function of cells [86].

It is noteworthy that inflammation is associated with hypoxia [87]. Infiltrating immune cells, invading pathogens, bacteria, and the increased energy demands of resident cells can force a sharp drop in available oxygen [5]. In colitis, hypoxia is observed in all parts of the mucosa [40]. These oxygen-limiting conditions impair β-oxidation, and can lead to remodelling of membrane lipids and proteins, with an increase in saturated fatty acids [29]. In addition, with pO_2_ reduction due to inflammation, IECs can release pro-inflammatory cytokines such as TNF. Furthermore, a drop in pO_2_ contributes to increased epithelial cell apoptosis [88].

The exact mechanism for the increase in hypoxia in IBD is not yet clear [40]; it probably occurs due to several factors. Inflammation may lead to an increase in oxygen consumption by epithelial cells or an increase in vasculitis (inflammation of blood vessel walls) and thus, a decrease in oxygen availability in the inflamed areas. It is also possible that during inflammation, migrating neutrophils consume local oxygen and thus, increase hypoxia in colitis [5,89].

In the inflamed mucosa, HIF-1α increases barrier protection genes, enhances innate immune responses, and activates an antimicrobial response by increasing β-defensins, antimicrobial peptides implicated in the resistance of the surface of epithelia to microbial colonization. β-defensin expression requires low-oxygen conditions and HIF-1α activation [90]. All of these activities of HIF-1α in colitis result in a protective response. On the contrary to HIF-1α, HIF-2α are essential for the maintenance of an epithelial inflammatory response and, when chronically activated, can also increase the pro-inflammatory response, intestinal injury, and cancer [40]. However, more work is needed to fully understand the dynamic regulation of HIF-1α and HIF-2α in inflamed mucosa.

IBD is classified into two broad groups: ulcerative colitis (UC) and Crohn’s disease (CD). In UC patients, inflammation is localised in the colonic region. In CD patients, inflammatory lesions are seen throughout the gastrointestinal tract, with some extra ulcerated mucosa and granulomas [91]. Both HIF-1α and HIF-2α expression are increased in the intestinal epithelium of UC and CD patients and in a mouse model of colitis [87].

The composition of the microbiota is frequently associated with IBD [92]. In IBD, there is an overall decrease in microbial diversity, including decreased abundance and even loss of major phyla of obligated anaerobe bacteria such as *Firmicutes* and *Bacteroidetes*, which, in healthy individuals, represent 90% of all microbes present [56,84,93,94]. This suggests the presence of elevated oxygen levels. Following this, the abundance of facultative anaerobic bacteria, such as Enterobacteriaceae, is markedly elevated in individuals with IBD. The abundance of facultative anaerobic bacteria in the colon is correlated with the distribution of oxygen emanating from the host tissue [56,84,95]. Whether oxygen pressure alterations are causative or a result of the dysbiosis found in IBD remains to be determined.

A homeostatic role for SCFA in the distal gut during inflammation is important in the progression of inflammatory diseases. It was reported that a reduction in SCFA is associated with IBD [63,92]. In addition, the epithelial transporter—the cell-specific butyrate transporter (encoded by SLC16A1)—is downregulated in the inflamed colonic mucosa of patients with IBD. The reduction in butyrate and subsequent inhibition of β-oxidation may be particularly detrimental in the context of intestinal inflammation [96,97]. Administration of exogenous butyrate promotes resistance to experimental colitis [98]. Butyrate can also protect mice from *Clostridium difficile*-induced colitis through a HIF-1-dependent mechanism [37]. SCFAs are currently the most studied bacterial metabolites that, from studies, can be identified as beneficial to colon homeostasis. IBD is correlated with disruption of commensal bacteria and the levels of expression of genes in commensal bacteria [99]. Although patients with IBD exhibit decreased overall diversity of the microbiome with alternation in microbiota, it is still not clear if these changes initiate disease pathogenesis or are secondary to inflammation [100]. The microbiota of patients with active CD is 50% less diverse and the microbiota of patients with UC is 30% less diverse [101]

Increased hypoxia, a lack of metabolites from the microbiota and a dysfunctional microbiota are the main characteristics of IBD. These characteristics of IBD underpin the disease course and contribute to extra-intestinal symptoms. Although of diagnostic value, which of the alterations are causative and which offer viable entry points for therapy beyond reducing symptoms remain to be investigated.

## 7. Conclusions

The differences in pO_2_ levels and energy demands of mucosal epithelial tissue in comparison to other tissues during physiological functions or disease are unique. Its study provides opportunities to understand intestinal metabolism in health and disease. There is a dynamic and fast fluctuation in cellular oxygen tension between the tip and base of the villus in the small intestine, where the base is better oxygenated and the tip of the villus is hypoxic. There is also a steep decrease in oxygen longitudinally from the small intestine towards the colon. The main part of the intestinal epithelium is constantly exposed to low oxygen, and this represents a prototypic environment in which adaptation to the hypoxia is a key. Hypoxia is regulated through HIF transcription factors that regulate the expression of 500 genes, among which many are crucial to mucosal barrier function, such as junctional proteins, antimicrobial proteins and mucus. In inflammatory bowel disease (IBD), this cellular oxygen tension in the intestine is dysregulated. Current understanding of IBD pathogenesis assumes that in genetically predisposed individuals, the interaction between microbiota and the mucosa immune system results in an inappropriate inflammatory response. However, the recurrence of the disease is triggered by environment agents that still need to be fully characterized. Future studies to better understand the equilibrium between microbial crosstalk, mucosal immune system and oxygen consumption are becoming increasingly recognized in the field of mucosal immunology and intestinal epithelial barrier biology and will help us to better understand how the epithelial barrier behaves in health and disease.

## Figures and Tables

**Figure 1 ijms-22-09170-f001:**
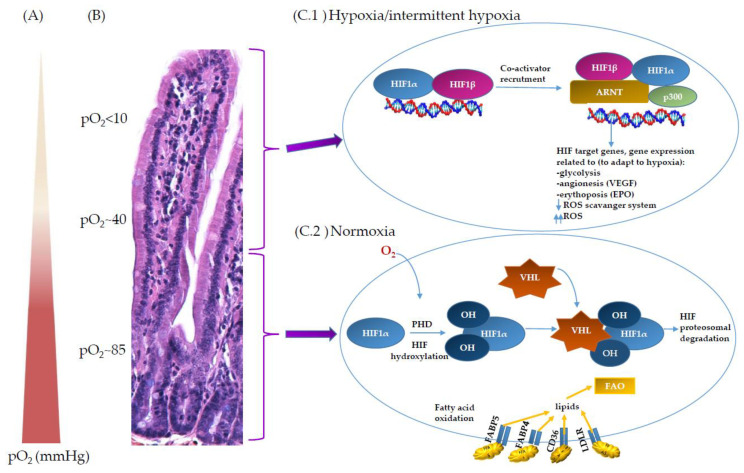
O_2_ and its regulation at the intestinal epithelial barrier site. (**A**) Counter current blood flow reduces local pO_2_ along the crypt–villus axis and results in low pO_2_ at the villus tip. (**B**) Histological section of small intestine; villus–crypt axis, BALB/cOlaHsd mouse. (**C**) Hypoxic/normoxic environment along crypt–villus axis. (**C.1**) Hypoxic condition or intermittent hypoxia at the upper part of the villa; inactive HIF hydroxylases. HIF-1 is composed of two subunits—oxygen-sensitive HIF-1α and HIF-1β. Due to low O_2_, PHD activity is decreased. HIF-1α is stabilised and binds to ARNT in the presence of co-activator p300; altogether, it activates transcription factor of HIF target genes. (**C.2**) Normoxic conditions at the bottom of the crypt. In the presence of O_2_, PHD enzyme hydroxylates the two proline residues on HIF, enabling the binding of VHL to the HIF subunit, which degrades HIF-1α subunits under normoxic conditions. Fatty acid oxidation (FAO) is predominately present in normoxic cells.

**Figure 2 ijms-22-09170-f002:**
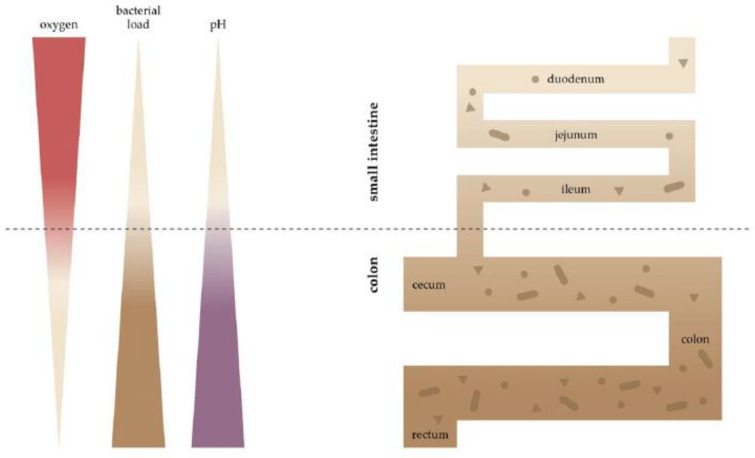
Longitudinal gradient of O_2_, bacterial load and pH along the intestine. Spatial heterogeneity of the gut microbiota in the gastrointestinal tract. The bacterial families of the small intestine and colon reflect physiological differences along the length of the gut. A gradient of oxygen and pH limits the bacterial density in the small intestine, whereas the colon carries high bacterial loads (darker brown = more bacteria).

**Figure 3 ijms-22-09170-f003:**
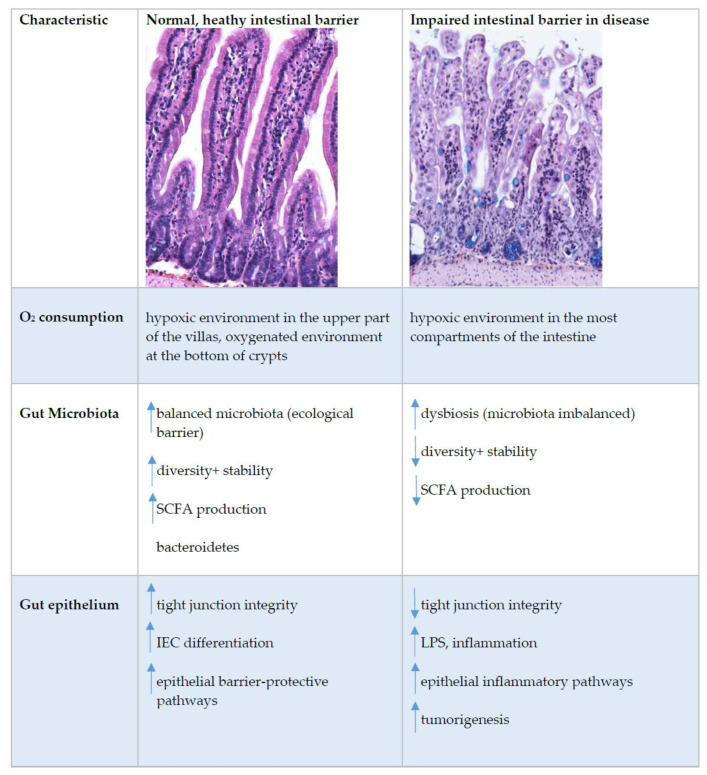
Intestinal epithelial barrier in health and disease. This figure shows immunohistochemistry picture of small intestine crypts–villus axis in healthy and impaired intestine in BALB/cOlaHsd mice and highlight the main characteristics where the differences at the healthy or impaired (in disease) intestinal barrier were detected. Up and down arrows indicate an increase and decrease, respectively. Abbreviations: IEC, intestinal epithelial cells; LPS, lipopolysaccharide; SCFA, short-chain fatty acids; TJ, tight junctions.

**Table 1 ijms-22-09170-t001:** The most important methods for quantitative measurement of the partial pressure of O_2_ in vivo.

Method	Approach	Advantages	Disadvantages	Reference
Clark electrode	current generated due to oxygen reduction at electrode (Pt) ^1^	simple; well established and documented; robust	not appropriate for low pO_2_ (consumes O_2_); not appropriate for large area; invasive, possible tissue damage; no spatial information	[18]
phosphorescence quenching	altered phosphorescence lifetime of porphyrin sensor as a function of oxygen concentration, detection using fibreoptic needle	not dependent on excitation light intensity; reduced background compared to luminescence methods; high spatial precision and accuracy; does not consume O_2_	invasive, possible tissue damage (but less than with electrodes); dye photobleaching	[19]
electron paramagnetic resonance (EPR)	altered relaxation rate of a spin probe due to presence of O_2_	very precise; non-invasive; 3D O_2_ profile	addition of exogenous paramagnetic substances (e.g., insoluble paramagnetic materials); probe may be endocytosed	[20,21]
magnetic resonance imaging (MRI)	^19^F or ^1^H relaxation rate in probe compound as a function of pO_2_	non-invasive; 3D O_2_ profile	expensive; probe compounds need to be administered at least several hours before measurement	[21,22,23]
positron emission tomography (PET), single-photon emission computed tomography (SPECT)	γ-rays (PET) or single γ-ray photons (SPECT) emitted by short-lived ^15^O	non-invasive; large measurement area or even full body; 3D O_2_ profile; possible to measure low pO_2_ values	expensive and experimentally demanding; safety concerns (γ-radiation)	[21,24]

^1^ Eppendorf polarographic microelectrode shares a similar principle.

## Data Availability

Not applicable.

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
