# Peer review of "Regulation of Oxygen Homeostasis at the Intestinal Epithelial Barrier Site"

_ijms, 2021, doi:10.3390/ijms22179170_

Round 1
Reviewer 1 Report
This review tackles different aspects of oxygen homeostasis in the intestinal epithelium. The review tackles an interesting topic, but ir is poorly written, giving an impression of unconnected ideas put together without a logical flow. Some sections (for example section 2) are an adapted copy of full section in previous review (i.e. section "Oxygen Landscape of the Intestine" in the review "Physiologic hypoxia and oxygen homeostasis in the healthy intestine. A Review in the Theme: Cellular Responses to Hypoxia", by Zheng et al. without contributing new information. Moreover, the english needs to be reviewed and edited. There are also repeated sentences (i.e. line 75). In summary, even though the topic is interesting, this review needs to be thoroughly reviewed by the author in order to improve language, clarity, coherence and critical analysis on the topic before being acceptable for publication in IJMS.
Author Response
We would like to thank the reviewer for the helpful comments. We have taken all advice on-board, improving the flow, depth and style of the review. We hope we have addressed all the concerns.
We improved and rewrote the individual chapters as suggested, taking care to reduce repeats and improving connection between chapters with small conclusion at the end of each chapter. We completely or partially rewrote most of the chapters, so the narrative is more logical and focused.
Reviewer 2 Report
Even though this manuscript covers an interesting topic, that is worth follow up, but the mauscript presented needs major reworking and improvements.
It needs major rewriting both in language, terminology and flow of argumentation. Multiple inconsistencies and inaccuracies need to be corrected.
Very often it seems mere listing of possibilies. It is either up or down in the same sentence, and often both options are indiscemately labelled as negative/pathological, despite the main topic of the manuscript is the occurence and maintainance of gradients (as physiological and functionally necessary).
Use of English language e. g. terms need to be checked throughout the manuscript.
It would help the reader to comprehend the various oxygen gradients (apparently across the epithelial cells, along the crypts (Figure 1, lines 93-94) and along intestine as an organ(lines 89-93), would be all separately graphically depicted (similar and in addition to Figure 1) and than compared.
Even though there are 3 references from 2019/20, most references are older than 5 years. It would enhance the manuscript if the authors would include a statement on which topics recent literature is missing. In several instances, e. g. the ‚HIF‘ complex, there is a multitude of ‚really‘ recent papers. Based on the recent surge in microbiota research in various medical fields, references from 2014 are not ‚state of the art‘.
Please clarify, the difference between ‚O2‘ and ‚pO2‘.
How does mentioning pH gradients connect to oxygen gradients in line 92?
Line 41: ‚at the baseline of the IEC lining…‘ does that refer tot he basal side (in contrast to apical side) of the IEC?
Oxygen Measuring Methods should be a separate chapter including a table and references.
Reference 5 deals largely with hypoxia (and inflammation), but not really with ‚a quick adaptation of oxygen in/due to blood flow‘. Please check this.
The title of reference 7 reads: ‚Mitochondria control…‘, while the text (line 45) states: ‚can mimic‘ please clarify this discrepancy, including the term: ‚mitochondrial signals represented by‘.Are metabolites mitochondrial signals? Or are there other not named mitochondrial signals? Compare to statement in lines 116-117.
Introduction of abbreviations is mainly done correctly, but missing for ROS (lines 45-46)
Lines ~70-76 duplication of content
Line 114 Mitochondria ‚are‘ (not is)
Figure 1: (tans-) location fo HIF is missing.
Lines 127/128: Twice the same process/result for ‚loosing‘ and ‚picking up‘ an electron?
Lines 206-208: How do anaerobic bacteria decrease the O2 content – per definition they do not consume O2 – it may even be poisonous to them. Or are facultative anaerobic bacteria ment?
Line 206: ‚Anareobic bacteria‘ are not ‚a species‘.
Author Response
We would like to thank the reviewer for the accurate and helpful evaluation of our manuscript. We tried to follow all advice suggested. We payed particular attention to your concerns and we rewrote the manuscript so that language, terminology and flow of argumentation is more consistent, without unnecessary repeats. We tried to address also individual question that you posed. We hope we explained it well and that you find the corrected and improved text sufficiently improved.
Even though this manuscript covers an interesting topic, that is worth follow up, but the mauscript presented needs major reworking and improvements.
It needs major rewriting both in language, terminology and flow of argumentation. Multiple inconsistencies and inaccuracies need to be corrected.
Very often it seems mere listing of possibilies. It is either up or down in the same sentence, and often both options are indiscemately labelled as negative/pathological, despite the main topic of the manuscript is the occurence and maintainance of gradients (as physiological and functionally necessary).
Use of English language e. g. terms need to be checked throughout the manuscript.
It would help the reader to comprehend the various oxygen gradients (apparently across the epithelial cells, along the crypts (Figure 1, lines 93-94) and along intestine as an organ(lines 89-93), would be all separately graphically depicted (similar and in addition to Figure 1) and than compared. The chapter 2 Oxygen scenery at the intestinal epithelial barrier has been rewritten. We agree with the suggestion of an additional figure. In Figure 2, we now show the gradient of oxygen, bacterial load and pH along the intestine as suggested.
Even though there are 3 references from 2019/20, most references are older than 5 years. It would enhance the manuscript if the authors would include a statement on which topics recent literature is missing. In several instances, e. g. the ‚HIF‘ complex, there is a multitude of ‚really‘ recent papers. Based on the recent surge in microbiota research in various medical fields, references from 2014 are not ‚state of the art‘
We tried to address your question and improved the citations used. We added additional new references from 2016-2021 where we thought it of particular relevance. From 101 papers citied in the latest version of the manuscript, 44,5% is from 2016-2021 and from 2019-2021 there are 17 citations now.
Please clarify, the difference between ‚O2‘ and ‚pO2‘.
We use O2 to indicate proportion of oxygen gas in the environment (exp. intestine). Although frequently measured as oxygen pressure, it is often evaluation on the % of total gas in the environment concerned. We have taken greater care to use pressure, mmHg, or proportion.
How does mentioning pH gradients connect to oxygen gradients in line 92? Chapter 2; Oxygen scenery at the intestinal epithelial barrier, has been completely rewritten. We removed the comparison with pH since this comparison is not the focus of the review.
Line 41: ‚at the baseline of the IEC lining…‘ does that refer tot he basal side (in contrast to apical side)
We apologise for the mistake, it should have been the baseline of epithelial cells lining the mucosa, where relatively low pO2 exists, referred to as physiological hypoxia. We have corrected this.
Oxygen Measuring Methods should be a separate chapter including a table and references. We have included a table for the different methods to assess O2 (Table 2). We think this would be of benefit to the readers.
Reference 5 deals largely with hypoxia (and inflammation), but not really with ‚a quick adaptation of oxygen in/due to blood flow‘. Please check this. Here we agree the main topic of the citied article (reference 5) is inflammation and not adaptation of oxygen levels, but hypoxia it is still part of the topic of the reference 5.
The title of reference 7 reads: ‚Mitochondria control…‘, while the text (line 45) states: ‚can mimic‘ please clarify this discrepancy, including the term: ‚mitochondrial signals represented by‘.Are metabolites mitochondrial signals? Or are there other not named mitochondrial signals? Compare to statement in lines 116-117. Since this text is predominately rewritten in the line 5 we will try to answer the exact question of the line 5 in the first version of the paper. The main concern here as we understand is this question concerning definition of ROS and metabolites. In this review, ROS is defined as mitochondrial-derived signal. While metabolite is meant as more general, it could be a derivative from the microbiota or metabolite from the TCA cycle produced in the mitochondria.
Introduction of abbreviations is mainly done correctly, but missing for ROS (lines 45-46) I added in brackets (ROS)
Lines ~70-76 duplication of content I added in brackets (ROS)
Line 114 Mitochondria ‚are‘ (not is) corrected
Figure 1: (tans-) location fo HIF is missing. Hopefully properly corrected in the picture for Figure 1
Lines 127/128: Twice the same process/result for ‚loosing‘ and ‚picking up‘ an electron? We removed the sentenced picked up electron, I agree it is not needed.
Lines 206-208: How do anaerobic bacteria decrease the O2 content – per definition they do not consume O2 – it may even be poisonous to them. Or are facultative anaerobic bacteria ment? We appologies for our mistake, we meant facultative anaerobic bacteria, which we now corrected.
Line 206: ‚Anareobic bacteria‘ are not ‚a species‘. Corrected
Expression AXIS we left in the text, since many times in paper it appears in combination as villa-crypts axis

Round 2
Reviewer 2 Report
This manuscript now nicely reviews the role of oxygen gradients in the intestine with regard to gut microbiota, IBD and enters into the underlaying signalling pathways. It has been majorly improved in language, readability and more recent literature has been included.
Nevertheless a few relatively minor issues remain, before this manuscript should be considered for publication:
- A thorough spell check is still needed (mainly in Table 2 and the acknowledgements; Lines 189,215,328,376,391,449)
- Naming those 4 proteins as ‚biomarkers‘ for hypoxia is somewhat superficial, even though the genes are hypoxia inducable (besides many others).(Lines 123-125)
- Please clarify (and correct) if ARNT is another name for HIF-1Beta (Lines 138/139) or a separate subunit//protein (as in Figure 1).
- Figure 1 would profit from a depiction of the nucleus (vs cytosolic compartment) – especially since HIF1-alpha localisation is important for both ist function and analysis by immunohistochemistry (as already mentioned in the text)
- It seems that both in line of argumentation and sequence of first mentionning in the text, numbering of Figures 1 and 2 should be exchanged.
- Oxygen measuring methods (in Table 2) should include more indepth details e. g. names of substances that need to be applied (injected). An important information would be spacial (and if applicable/available) temporal resolution – expecially since both mouse and human have different requirements for analysis.
- This manuscript is a review not a ‚study‘ (Line 420)